# Lipopolysaccharide triggers different transcriptional signatures in taurine and indicine cattle macrophages: Reactive oxygen species and potential outcomes to the development of immune response to infections

**Raquel Morais de Paiva Daibert**[1�], **Carlos Alberto Oliveira de Biagi Junior**[2‡], **Felipe de Oliveira Vieira**[1‡], **Marcos Vinicius Gualberto Barbosa da Silva**[3‡], **Eugenio Damaceno Hottz**[1‡], **Mariana Brandi Mendonça Pinheiro**[1‡], **Daniele Ribeiro de Lima Reis Faza**[3‡], **Hyago Passe Pereira**[1‡], **Marta Fonseca Martins**[3‡], **Humberto de Mello Brandão**[3‡], **Marco Antônio Machado**[3�উ], **Wanessa Araújo Carvalho**[3]*

1 Federal University of Juiz de Fora, Juiz de Fora, Minas Gerais, Brazil, 2 Ribeirão Preto Medical School, University of São Paulo, Ribeirão Preto, Brazil, 3 Embrapa–Dairy Cattle Research Centre, Juiz de Fora, Minas Gerais, Brazil

উ These authors contributed equally to this work.
‡ These authors also contributed equally to this work.
* wanessa.carvalho@embrapa.br

## Abstract

Macrophages are classified upon activation as classical activated M1 and M2 anti-inflammatory regulatory populations. This macrophage polarization is well characterized in humans and mice, but M1/M2 profile in cattle has been far less explored. *Bos primigenius* taurus (taurine) and *Bos primigenius indicus* (indicine) cattle display contrasting levels of resistance to infection and parasitic diseases such as C57BL/6J and Balb/c murine experimental models of parasite infection outcomes based on genetic background. Thus, we investigated the differential gene expression profile of unstimulated and LPS stimulated monocyte-derived macrophages (MDMs) from Holstein (taurine) and Gir (indicine) breeds using RNA sequencing methodology. For unstimulated MDMs, the contrast between Holstein and Gir breeds identified 163 Differentially Expressed Genes (DEGs) highlighting the higher expression of C-C chemokine receptor type five (*CCR5*) and *BOLA-DQ* genes in Gir animals. LPS-stimulated MDMs from Gir and Holstein animals displayed 1,257 DEGs enriched for cell adhesion and inflammatory responses. Gir MDMs cells displayed a higher expression of M1 related genes like *Nitric Oxide Synthase 2* (*NOS2*), *Toll like receptor 4* (*TLR4*), *Nuclear factor NF-kappa-B 2* (*NFKB2*) in addition to higher levels of transcripts for proinflammatory cytokines, chemokines, complement factors and the acute phase protein Serum Amyloid A (SAA). We also showed that gene expression of inflammatory M1 population markers, complement and SAA genes was higher in Gir in buffy coat peripheral cells in addition to nitric oxide concentration in MDMs supernatant and animal serum. Co-expression analyses revealed that Holstein and

**Data Availability Statement:** All RNA sequence datafiles are available from the GEO repository database (https://www.ncbi.nlm.nih.gov/geo; accession number GSE 147813) open for public access.

**Funding:** This research was supported by grants from the National Council for Scientific and Technological Development (CNPq; WAC 471864/2013-7 and MAM 472578/2013-8; http://www.cnpq.br). RMPD was supported by Coordination for the Improvement of Higher Education Personnel (CAPES; HMB, MAM and WAC grant from 8888.159607/2017-1; https://www.capes.gov.br) and National Institute of Science and Technology - Animal Science (INCT-CA; 372348/2019-0; http://inct.cnpq.br).

**Competing interests:** The authors have declared that no competing interests exist.

Gir animals showed different transcriptional signatures in the MDMs response to LPS that impact on cell cycle regulation, leukocyte migration and extracellular matrix organization biological processes. Overall, the results suggest that Gir animals show a natural propensity to generate a more pronounced M1 inflammatory response than Holstein, which might account for a faster immune response favouring resistance to many infection diseases.

## Introduction

The modern domestic cattle is composed by two distinct subspecies, *Bos primigenius taurus* (taurine cattle) and *Bos primigenius indicus* (indicine cattle), originated from European and Asian continent, respectively [1]. These subspecies show many differences in morphophysiological and genetic parameters [2, 3] which influence infection and parasitic diseases outcome in bovine cattle [3–5]. Even within subspecies, breeds show differences in the amount and response of immune cells as well as inflammatory mediators production [6–9]. In this context, researchers have highlighted the genomic background impact on *in vivo* proinflammatory innate immune, metabolic and endocrine responses to bacterial lipopolysaccharide (LPS) between two taurine breeds [10]. LPS is the main cell wall component of Gram-negative bacteria that triggers the production and release of endogen mediators including platelet-activating factors and thromboxanes, reactive oxygen species as nitric oxide, interleukin-1 (IL-1), IL-6 and tumour necrosis factor alpha (TNFα) from monocytes/macrophages of host vertebrate species [11, 12]. LPS activates cellular responses by association to TLR4 membrane receptor and CD14 co-receptor through recruiting adaptor molecules and culminating in activation of proinflammatory transcription factors [13]. The inflammatory innate immune response is mainly mediated by monocytes, macrophages and neutrophils which recognize pathogen-associated molecular patterns, such as LPS. These cells will phagocyte and kill pathogens and simultaneously coordinate T helper and memory immune response development by synthesizing inflammatory mediators and cytokines [14].

Bovine monocytes and macrophages show divergent subpopulations which seems to have common characteristics with humans and murine models [15]. Macrophages subpopulations are characterized as proinflammatory, classically activated M1 and anti-inflammatory, regulatory M2 populations [16]. The M1/M2 macrophage polarization nomenclature was introduced in the year 2000, based on the propensity of C57BL/6J macrophages to be more easily activated to produce NO (M1 polarized) than Balb/c mice (M2 polarized) [17] which mediate differences in susceptibility to a variety of infection diseases [18]. The macrophage polarization phenotypes have been well characterized in humans and mice, but M1/M2 macrophage profiling in cattle has been far less explored. Since taurine and indicine cattle breeds show different levels of resistance to infection and parasitic diseases, we hypothesized that these phenotypes may be related to specific macrophage activation pathways associated to type 1 and 2 immune response in cattle. Therefore, the aim of this study was to check if the MDMs from taurine (Holstein) and indicine (Gir) breeds could exhibit different transcriptional signatures triggered by LPS stimulation that might affect innate immune activation influencing the outcome of parasitism and infection in cattle.

## Materials and methods

### Animals

Indicine (Gir; n = 6) and taurine (Holstein; n = 6) animals aged from 6 to 12 months old were produced at Embrapa Dairy Cattle Research Centre experimental station in Coronel Pacheco,

Brazil. All animals were healthy, vaccinated accordingly, kept stabled and *ad libitum* fed for three months prior to sample collection in order to certify they were free from any chemical agent, infections and parasitic diseases that could interfere with the trials. All animals were housed to be used in further research after this experimental trial was finished. The experimental design was approved by Embrapa Dairy Cattle Research Centre Ethics Committee filed under CEUA 5578010817.

## Blood collection and differentiation of bovine monocyte-derived macrophages (MDMs) *in vitro*

Peripheral blood was individually collected to obtain monocytes which were *in vitro* differentiated into macrophages as already described elsewhere with minor adjustments in the original protocol [19]. Fetal bovine serum (FBS) was used to avoid interference in the cellular differentiation caused by individual biochemical components present in autologous serum. Briefly, leukoplatelet layer was separated from 60ml of peripheral blood samples by centrifugation at 300 xg for 10 min, followed by suspension in 5ml of phosphate-buffered saline (PBS). The mononuclear cells were separated by hydrophilic density polysaccharide gradient 1,077 g/ml Ficoll (GE Healthcare, Chicago, USA) by centrifugation at 400xg for 40 minutes at room temperature. The mononuclear cells layer was suspended in RPMI-1640 medium (Sigma-Aldrich, Saint Louis, USA) supplemented with inactivated 10% FBS (LGC Biotecnologia, Cotia, Brazil), 2 mM L-glutamine (Sigma-Aldrich, Saint Louis, USA), 10 mM sodium pyruvate (Sigma-Aldrich, Saint Louis, USA), non-essential amino acid solution 1% MEM (Sigma-Aldrich, Saint Louis, USA) and 1% antibiotic antimycotic solution (Sigma-Aldrich, Saint Louis, USA). Adherent cells were differentiated into macrophages at cell chambers with 5% $CO_2$ at 37˚C for 11 days, as already described in literature [19].

## *In vitro* characterization of bovine macrophage differentiation from monocytes

Bovine macrophage differentiation from adherent mononuclear cells were characterized by morphological changes followed by light microscopy (Zeiss, Oberkochen, Germany) and cell immunophenotyping. Flow cytometry was used to quantify CD14 and CD11b expression on mononuclear cell surface and evaluate adherent cell differentiation at 24h and 11 days of cell culture. For that, adherent cells were collected with cell dissociation solution non-enzymatic (Sigma-Aldrich, Saint Louis, USA) as described by the manufacturer's recommendations. The viable cell count was performed by Trypan blue exclusion test [20]. A total of $2x10^5$ cells were isolated and each sample marked with anti-CD-14-FITC (Bio-Rad, Hercules, USA) and anti-CD11b-FITC (Bio-Rad, Hercules, USA) antibodies individually. Mouse IgG1 and IgG2 (Bio-Rad, Hercules, USA) were used as isotype controls according to manufacturer's recommendations. After incubation with detection antibodies for 30 min, cells were washed with PBS (Sigma-Aldrich, Saint Louis, USA) and acquired with FacsVerse cytometer (BD Biosciences, Franklin, USA). FlowJo software (Tree Star Inc., Ashland, USA) was used to quantify the percentage of mononuclear cells expressing CD11b and CD14 on 24 hours and 11 days of cell culture differentiation. Statistical analyses were performed by GraphPad Prism software version 5.0 for Windows (GraphPad Software Inc, San Diego, USA), adopting significance of P<0.05.

## Bovine MDM LPS stimulation, library preparation and RNA sequencing

After 11 days of *in vitro* cell differentiation, taurine (Holstein; n = 6) and indicine (Gir; n = 6) MDMs were individually incubated for 48h with 100 ng/ml LPS from *Escherichia coli* O111:B4

(Sigma-Aldrich, Saint Louis, USA). For the negative control without stimulation, only culture medium was used. The total RNA from cells was extracted with RNeasy Micro kit (Qiagen, Hilden, Germany) according to manufacturer's instructions. RNA samples were quantified with Nanodrop 1000 (Thermo Scientific, Waltham, USA) spectrophotometer and integrity evaluated by Bioanalyzer using RNA 6000 Pico kit (Agilent Technologies, Santa Clara, USA) according to manufacturer's instructions.

TruSeq Stranded mRNA Sample Preparation Kit (Illumina, San Diego, USA) was used to generate cDNA libraries from samples that showed a minimum of 100 ng total RNA and RNA integrity number (RIN) over 7,00. RNA sequencing was performed in 16 MDM samples using the HiSeq 2500 DNA sequencer (Illumina, San Diego, USA) using the HiSeq SBS Kit v4 (Illumina, San Diego, USA), according to manufacturer's recommendations for gene expression profiling experiments focusing a quick snapshot of highly expressed genes. Unstimulated (n = 8) and LPS treated MDM (n = 8) RNA samples in which half were from Gir and half from Holstein animals were sequenced generating a depth of 10 million 100 bp paired-end reads per sample. NGS data were deposited at GEO repository (https://www.ncbi.nlm.nih.gov/geo) on query GSE147813.

The quality of RNA sequencing reads was verified with FastQC software v0.11.7 (https://www.bioinformatics.babraham.ac.uk/projects/fastqc/). After that, reads were mapped to the bovine reference genome ARS-UCD1.2 from Ensembl database (Bos_taurus.ARS-UCD1.2.dna.toplevel.fa.gz). Spliced Transcripts Alignment were accomplished using the Spliced Transcripts Alignment to a Reference (STAR) software v.2.6.0c [21] using the annotation archive from the same database (Bos_taurus.ARS-UCD1.2.101.gtf.gz). Only exclusively mapped reads were considered for MDMs transcriptome analysis. Estimation of transcript abundance was accomplished with HTSeq-count software v0.10.0 [22].

## Differentially expressed genes (DEGs) and enrichment analysis

The contrast of transcript abundance between Holstein and Gir breeds for each unstimulated and LPS treated MDMs was performed by the EdgeR package version 3.8 [23] and R version 3.5.0 (http://www.R-project.org). Briefly, gene counts for each contrast was submitted to an initial filtering step, including genes with at least one count per million (CPM) in at least four libraries. The differentially expressed genes (DEGs) were considered statistically significant when false discovery rate (FDR) was <0.05 and Log of fold change (LogFC) were ≥1 in paired comparison. An interactive Venn diagram viewer (Jvenn, http://jvenn.toulouse.inra.fr/app/index.html) [24] was used to determine shared expression data between breeds and stimuli. Heatmaps were elaborated for breed contrasts using the Heatmapper software (http://www2.heatmapper.ca/expression/).

The Database for Annotation, Visualization and Integrated Discovery version 6.8 (DAVID, http://david.abcc.ncifcrf.gov/) [25] was used for DEGs functional annotations, for obtaining official gene symbols and for ontology analyses (GO) of DEGs (LogFC≥1; CPM>1; FDR<0.05).

## Co-expression analysis using Regulatory Impact Factors (RIF) and Partial Correlation and Information Theory (PCIT)

The transcript abundance counting tables was used as input to CeTF [26, 27] package in R to run the co-expression of bovine Transcription Factors using RIF [28] and PCIT [29] analyses. This package was also used to obtain the ontologies related to Biological Processes from the Gene Ontology database [30, 31] associated to the bovine key transcription factors (KeyTF) [32] and DEGs in context of metabolic pathways. The Cytoscape software [33] was used to

visualize and manipulate the interaction among DEGs, KeyTF and enriched biological process networks. The Diffany plugin [34] was used to infer differential molecular networks between Holstein and Gir MDMs stimulated with LPS.

## Quantitative real-time-PCR (RT-qPCR)

The RT-qPCR was performed to validate RNA-Seq data analyses in addition to investigate the buffy coat cells gene expression involved in the immune response of animals used in the trial. RNA samples were extracted from both MDMs and peripheral buffy coat from Holstein (n = 6) and Gir (n = 6) samples. RNA extraction from unstimulated and LPS treated MDMs from Gir (n = 4 per stimulus) and Holstein (n = 4 per stimulus) animals was performed with RNeasy Micro kit (Qiagen, Hilden, Germany) according to the manufacturer's instructions. The buffy coat cells were obtained from each animal by peripheral blood centrifugation (300xg). The white cell layer was removed followed by ACK (Ammonium-Chloride-Potassium) red cell lysis and RNA extraction using the SV Total RNA Isolation System (Promega, Madison, USA) according to the manufacturer's instructions. The total RNA extracted was quantified by the Nanodrop 1000 Spectrophotometer (Thermo Scientific, Waltham, USA) and its integrity was evaluated by Bioanalyzer with RNA 6000 Pico kit (Agilent Technologies, Santa Clara, USA). Samples were submitted to cDNA synthesis by SuperScript IV First-Strand Synthesis System kit (Thermo Scientific, Waltham, USA), according to manufacturer's instructions.

RT-qPCR assays were performed with PowerUp SYBR Green Master Mix (Thermo Scientific, Waltham, USA), according to the manufacturer instructions, using the 7500 Fast Real-Time PCR System (Thermo Scientific, Waltham, USA). Gene amplification targets were selected after RNA sequencing analysis based on DEGs enriched for immunological process and M1/M2 population biomarkers. Primers sequences were obtained using Primer Express v.3.0 software (Applied Biosystems, Foster City, California, EUA) and described in the S1 Table. The RT-qPCR efficiency stablished for all gene targets ranged from 95–105%. *Ribosomal Protein Lateral Stalk Subunit P0* (*RPLP0*) and *Ubiquitin* genes were used as reference based on expression stability calculated according to GeNORM procedure [35]. Average of Ct values from targets and reference genes were calculated for each sample using ABI Real Time PCR 7500 software v2.3 (Thermo Scientific, Waltham, USA). Statistical analyses were performed using the SigmaPlot 11.0 (Systat Software Inc., San Jose, USA) to test the equality between relative gene expression variation means from breeds and treatments. We adopted P<0.05 as significant threshold for the differences resulted from experimental contrasts. The graphical representation was performed by GraphPad Prism software version 5.0 (GraphPad Software, San Diego, USA).

## Nitric oxide (NO) dosage

The NO production was indirectly detected in MDMs culture supernatants and serum from Holstein and Gir samples by quantifying the NO breakdown product nitrite using Griess method [36]. Briefly, nitrite was quantified by reaction with 0.5% sulfanilamide (Sigma-Aldrich, S0251) and 0.05% N-(1-Naphthyl)-ethylenediamine dihydrochloride (Sigma-Aldrich, N-9125). A standard curve was prepared by serial dilution of sodium nitrite (Sigma-Aldrich, S2252). The absorbances of Griess reactions were determined by SpectraMax microplate reader (Molecular Devices, San Jose, USA), using 595nm for each sample in duplicate. Means and standard deviation (SD) of Nitrite concentration (µM) were calculated for each sample by linear regression curves and used for statistical analyses performed by SigmaPlot version 11.0 (Systat Software Inc., San Jose, USA) adopting P<0.05 as significance threshold. Graphic

representation was obtained by GraphPad Prism version 5.0 (GraphPad Software, San Diego, USA).

## Results and discussion

### Bovine monocyte-derived macrophages (MDMs), LPS treatment and RNA sequencing

Regarding the MDM production, instead of using autologous serum [19], which contains cytokines and chemokines that vary individually and might affect macrophage activation phenotype, FBS was added to cell culture to assure assay standardization during MDM differentiation. Monocytes are the main adherent cells present in peripheral blood and express less CD14 and more CD11b receptors on cellular plasmatic membrane (CD14$^{LOW}$CD11b$^{HIGH}$) while macrophages show the opposite expression pattern (CD14$^{HIGH}$CD11b$^{LOW}$) [37, 38]. For human and murine species, MDM shows low CD14 expression, which is increased after days of *in vitro* differentiation [39]. Corroborating these findings, our flow cytometry analysis of bovine adherent mononuclear cells showed an increase of CD14 receptor expression whereas CD11b decreased after 11 days of cell differentiation (P = 0.015; Fig 1A and 1B). Indeed, the bovine adherent mononuclear cells, characterized as monocytes (CD14$^{LOW}$CD11b$^{HIGH}$), displayed morphological microscopic changes on days 1, 5, 8 and 11 of differentiation (Fig 1C) showing irregular spreading and elongation compatible to macrophage traits [19, 40]. Once bovine MDM (CD14$^{HIGH}$CD11b$^{LOW}$) was obtained from Holstein (taurine) and Gir (indicine) blood samples, LPS was used to access the *in vitro* inflammatory immune response pattern displayed by these cattle breeds, using RNA sequencing.

The quality control of the sequenced reads showed Phred scores over 32, sequence length around 100 base pairs (bp), total of reads over 10 million per sample, and percentage of deduplicated reads around 46% (Table 1). The average number of reads mapped to reference

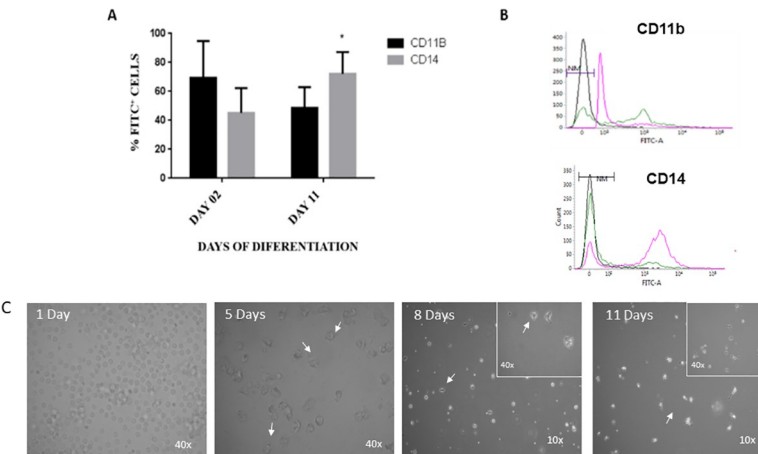

**Fig 1.** *In vitro* **differentiation of bovine monocytes (CD11b$^{hi}$CD14$^{low}$) into macrophages (CD11b $^{low}$CD14$^{hi}$) characterized by flow cytometry and light microscopy.** (A) Percentage of mononuclear cells expressing CD11b (black boxes) and CD14 (gray boxes) receptors on cellular membrane on days 2 and 11 after *in vitro* macrophage differentiation (n = 6; average ± SD; *P = 0.015). (B) Representative overlapped histograms displaying fluorescence intensity on FITC channel unstained cells (NM; black line) and individual acquired CD11b FITC and CD14 FITC stained cells on day 2 (green line) and day11 (pink line) after monocyte isolation and differentiation into macrophages. (C) Typical photos of bovine monocyte differentiation into macrophages taken by optical microscopy showing its original magnification (x10 or x40) and the number of days in cell culture (1, 5, 8 and 11 days). White arrows indicate cells with adherence to the flasks at day 5; cells spreading and acquiring primary shape of macrophage at day 8; and final differentiation into macrophage at day 11 showing elongation and spreading patterns.

**Table 1. Quantity and quality of RNA sequence reads obtained from Gir and Holstein MDMs.**

| Treatment | Breed | Total sequences | Phred Quality Score | Sequences remaining if deduplicated (% ± SD) |
|---|---|---|---|---|
| LPS | Gir | 10,385,461 ± 839,114 | > 32 | 44,74 ± 1,76 |
| LPS | Holstein | 10,894,776 ± 1,322,852 | > 32 | 46,01 ± 2,88 |
| Unstimulated | Gir | 10,627,817 ± 1,235,564 | > 32 | 46,76 ± 2,09 |
| Unstimulated | Holstein | 10,773,240 ± 381,359 | > 32 | 47,94 ± 1,94 |

Total number and quality evaluated by FastQC v0.11.7 software from *in vitro* unstimulated and LPS treated (100ng/ml) MDMs of Holstein and Gir cattle breeds.

genome ARS-UCD1.2 per sample (R1+R2) was 9.68±0.96 million depth, and the average percentage of uniquely mapped reads of 90.67%±1,20 (S1 Fig). NGS representativeness also showed a similar number of transcripts detected for each treatment in both breeds (CPM>1.0; S1 Fig). Overall, these results indicate that RNA sequences generated displayed high-quality scores and levels of redundancy ensuring that read abundance was not affected by biases during the library construction. Our experiments used a low depth [41] but enough to display a high coverage, which allowed to access the major genes involved in LPS response in Gir and Holstein breeds.

## Breed-specific bovine MDM transcriptional signatures and its impact on inflammation development for Gir and Holstein cattle

In order to access the *in vitro* major key process triggered by LPS in MDMs from Gir and Holstein animals, four different contrasts were used to generate lists of Differentially Expressed Genes (DEGs): (i) unstimulated and LPS treated MDMs within the same breed–Holstein (S2 Table) and Gir (S3 Table); (ii) unstimulated and LPS treated MDMs from Gir and Holstein (S4 Table) and (iii) LPS treated MDMs from Holstein and Gir breeds (S5 Table). Although MDMs were obtained under the same cell culture parameters, different numbers of DEGs were found in each analysed contrast, especially the ones that compare Gir and Holstein samples (Fig 2A). A total of 955 exclusive DEGs were found in the contrast between LPS treated MDMs from Holstein and Gir breeds, which accounted for 96.4% of the total unique genes in each contrast (Fig 2B). These results suggest that Gir MDMs are more responsive to LPS in comparison to Holstein samples and this might account for different outcome of inflammatory response among these breeds.

The Heatmap of RNA sequencing data showed different patterns of gene expression according to the bovine genetic background and MDM stimulus (Fig 3). In order to access the major biological processes that mediated the different gene expression patterns, DEGs were enriched for all analysed contrasts (S6–S9 Tables). DEGs enrichment analysis of unstimulated MDMs from Holstein and Gir returned no biological process showing FDR≤0.05 (S6 Table) although indicated putative differences in immune response development (P = 0.001 and FDR = 1.99). Gir MDMs showed higher expression of *Bovine Leukocyte Antigen* (*BOLA*) family (*BOLA-DQ A2*, logFC = -6.9 and P = 3.80E-10; *BOLA-DQA5*, logFC = -6.2 and P = 7.68E-08; *BOLA-DQB*, logFC = -3.5 and P = 0.0006; Fig 3A; S4 Table) and *C-C chemokine receptor type 5* (*CCR5*; logFC = -1.9 and P = 0.0006; Fig 3A; S4 Table). Interestingly, BoLA receptor family mediate antigen presentation to T-lymphocytes and its variant alleles has been linked to differences in resistance to many disease, including mastitis caused by gram negative bacteria [42–44]. The CCR5 receptor acts on leucocyte recruitment to inflammation site [45] and, coupled with BOLA, might influence differential outcomes of immune response in each bovine breed. DEGs enrichment from the contrast between unstimulated and LPS treated Holstein MDMs did not display biological processes showing FDR≤0.05 (S7 Table) although genes

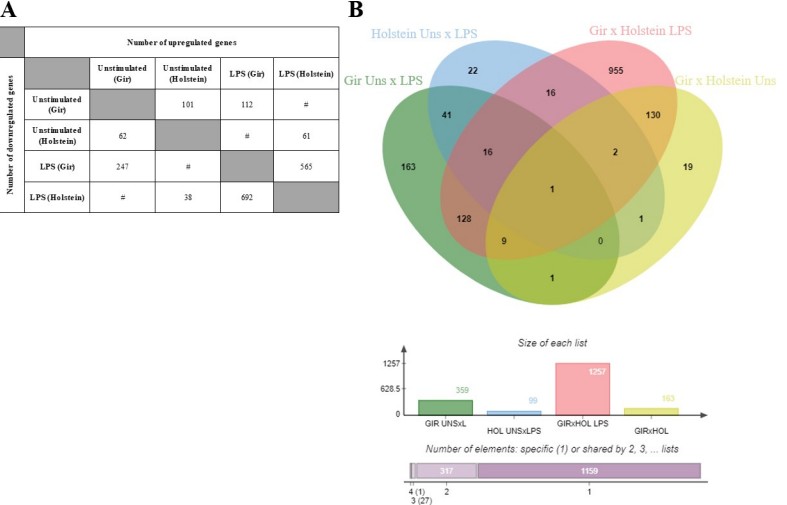

**Fig 2. Overview of transcription data from MDMs of Holstein and Gir breeds.** (A) Summary of numbers of upregulated and downregulated genes (FDR<0.05, LogFC≥1) determined by amounts of evaluated transcripts (CPM>1) between *in vitro* unstimulated and LPS (100 ng/ml) treated MDMs from Holstein (n = 4) and Gir (n = 4) MDMs. "#" denotes unrated contrasts. (B) Venn diagram showing DEGs (FDR<0.05, LogFC≥1) shared among all evaluated contrasts. Green diagram: unstimulated versus LPS treated Gir MDMs. Blue diagram: unstimulated cells versus LPS treated Holstein MDMs. Pink diagram: LPS treated Holstein versus Gir MDMs. Yellow diagram: unstimulated cells Holstein versus Gir MDMs. The graphic bars show the total number of DEGs for each contrast in the Venn diagram followed by numbers of specific genes in each contrast or shared between two, three or four contrasts.

involved in inflammatory response triggered by LPS were differentially expressed. It is noteworthy that *Nitric Oxide Synthase 2* (*NOS2*) was not found as DEG in contrasts between Holstein treated MDMs although the *Negative Regulator of Reactive Oxygen Species* (*NRROS*) was upregulated in LPS stimulus (logFC = -1.5 and P = 7.65E-05; S2 and S7 Tables). On the other hand, unstimulated vs LPS treated Gir MDMs displayed various DEGs enriched for chemokine-mediated signalling pathway (P = 1,32E-09 and FDR = 2,20E-06), cellular response to tumour necrosis factor (P = 2,45E-09 and FDR = 4,08E-06), cell chemotaxis (P = 2,84E-09 and FDR = 4,73E-06) and cellular response to interleukin-1 (IL-1; P = 5,13E-07 and FDR = 8,54E-04), along with additional biological processes (S8 Table). Unstimulated Gir MDMs showed higher levels of transcripts for *MHC* and *CCR5* in relation to Holstein animals, in addition to various DEGs enriched for biological process in unstimulated vs LPS treated Gir MDMs. This fact might account for a faster response against pathogens favouring resistance to diseases in indicine animals.

When we compare the transcriptome data from Holstein and Gir LPS activated MDMs, the differences on gene expression were augmented which culminated in very distinct transcriptional signatures (Fig 3B). Gene expression of *Toll-like receptor* (*TLR4*, logFC = -1.08 and P = 0.002) and *Nuclear factor-κB* (*NFκB*, logFC = -0.80 and P = 0.020) were higher in LPS stimulated MDMs from Gir than Holstein animals (S5 Table). Taking a deep look into the signalling pathways triggered by LPS, the Nuclear factor NF-kappa-B (NFκB) activation occurs, among other ways, through recognition of LPS by TLR4. The TLR4 and their co-receptor CD14 activate recruit adaptors molecules [46] which increase the complement 3 (C3) receptor expression [47] and culminate in Myeloid Differentiation primary response 88 (MyD88) and NFκB activation. The *complement factor 3* (*C3*, logFC = -1.28574 and P = 0.04) was also highly expressed in Gir LPS stimulated MDMs (Fig 3B; S5 Table) which could be involved in the indicine resistance to tick infestations and babesiosis [48], mastitis [49, 50] and heat stress [51].

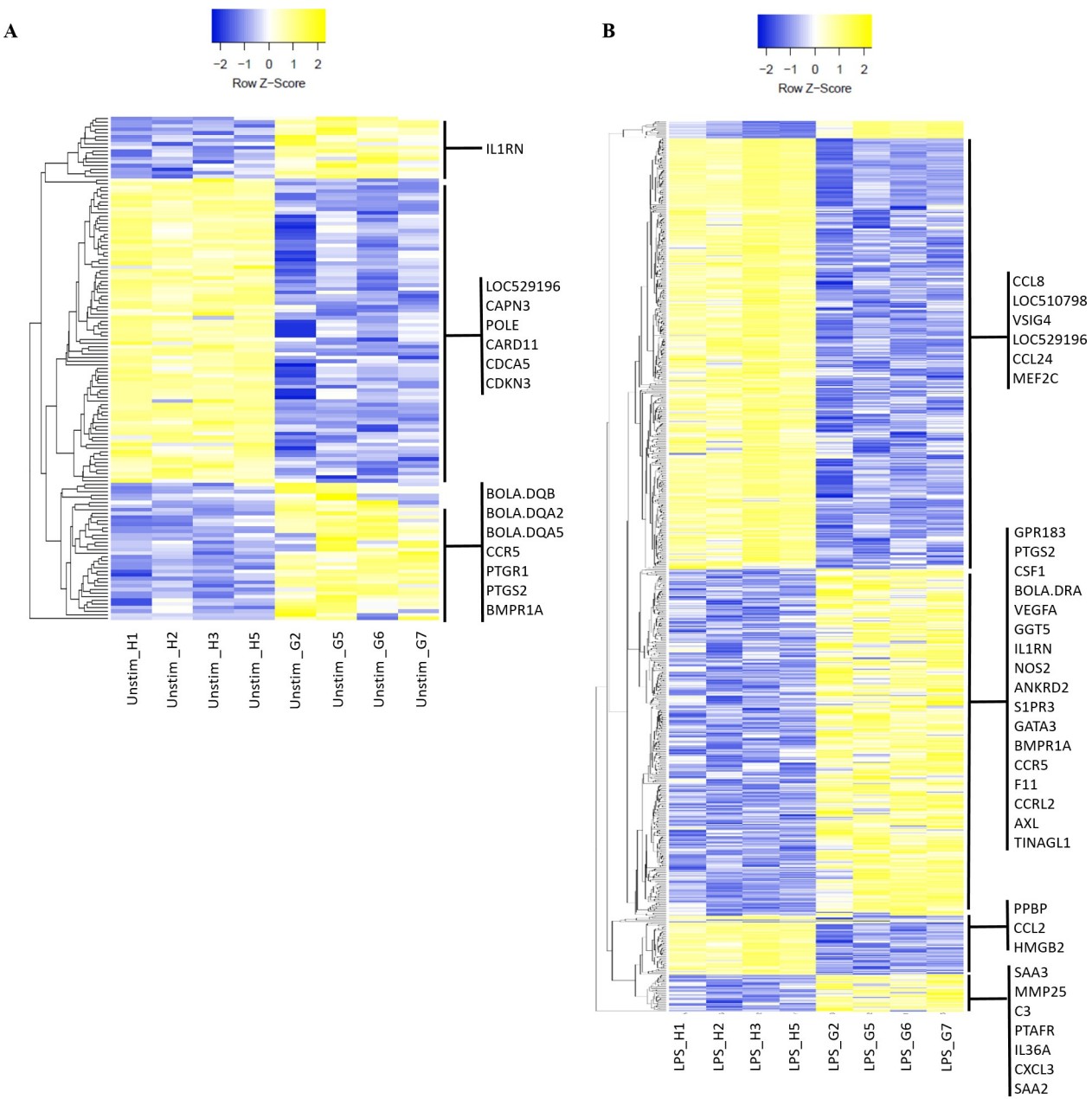

**Fig 3. *In* vitro stimulated MDMs transcriptional profile of Holstein and Gir breeds.** Heatmap of RNA sequencing data characterized as row-wise Z scores in CPM. Heatmap Z-scores were calculated for each row (each gene) and each column (each sample) and plotted according to the normalized expression values. (A) Unstimulated MDMs from Holstein (n = 4) and Gir (n = 4) breeds displayed 139 DEGs (FDR<0.05, LogFC≥1) related to biological processes of antigen processing and presentation via MHC class II, immune response and G1/S transition of mitotic cell cycle and (B) LPS stimulated MDMs from Holstein (n = 4) and Gir (n = 4) breeds displayed 920 total DEGs (FDR<0.05, LogFC≥1) related to biological processes of inflammatory response, regulation of cell proliferation and cell chemotaxis. The main DEGs were highlighted according to the enriched biological processes (P<0.05 and FDR≤0.05). H1, H2, H3, H5: Holstein samples; G2, G5, G6, G7: Gir samples.

The MyD88 dependent pathway activates NFκB and the pro-inflammatory cytokines production, promoting the recruitment and activation of Interleukin-1 Receptor Associated Kinases 1, 2 and 4 (IRAK1, IRAK2 and IRAK4), while the independent pathway of this molecule, via TIR-domain-containing adapter-inducing interferon-β (TRIF) induces type 1 IFNs [52]. It is noteworthy that *IRAK2* (logFC = -0.97 and P = 0.018) was found upregulated in Gir LPS

treated MDMs when compared to Holstein (S5 Table). These cascades result in many inflammation mediators biosynthesis, such as TNFα and IL-6 [53] that are associated to acute phase response which controls innate and adaptive immune response development. In addition, Gir MDMs also express more transcripts for the acute phase protein *Serum Amyloid A 2* and *3* (*SAA2* and *SAA3;* logFC = -7.01, P = 3.30E-07 and logFC = -1.78, P = 1.59E-06, respectively). The transcription factor NFκB can also be activated via nucleotide-binding oligomerization domain-like receptors (NOD-like receptors), independently of TLR [54]. The TLR signalling pathway and NOD-like receptor-associated inflammasome activation are required for active IL-1β secretion, which binds to its receptor IL-1R1 after caspase cleavage and activates NFκB [55]. The *IL1RN* gene encodes an IL-1 receptor antagonist protein (IL-1R1), which competes with IL-1 and inhibits the IL-1α e IL-1β synthesis [56]. These cytokines are endogenous pyrogen that control the inflammation development through modulation of cell surveillance, increasing expression of adhesion molecules and inducing secretion of acute phase proteins [57–62]. Interestingly, our results showed that *IL1RN* (logFC = -3.15 and P = 0.002), *IL1RL* (logFC = -6.02 and P = 4.19E-11), *IL36* (logFC = -1.51 and P = 1.39E-05), *NOD2* (logFC = -0.99 and P = 0.021) and *NFκB2* (*NFKB2;* logFC = -1.04 and P = 0.002) genes were highly expressed in Gir LPS stimulated MDMs (Fig 3B; S5 Table). Since these genes regulate tightly the cell activation and fate, DEG enrichment analysis of LPS treated MDMs from Holstein and Gir displayed many biological processes associated to cell division and replication (S9 Table). In this way, Gir MDMs appear to have a natural tendency to generate more pro-inflammatory immune response through increased activation and recruitment of leukocyte to the site of inflammation. Thus, it is reasonable to suggest more detailed studies in order to take a deeper look in the signalling pathways that underly phenotypes of inflammatory response regarding LPS activation in bovine macrophages and its influence on the outcome of inflammatory immune response to bacterial infections.

**Co-expression analysis.** Although transcription factors (TF) play a central regulatory role in cell biology, the detection of their expression in RNA sequence analyses is limited due to their low, and often sparse, expression. The partial correlation and information theory approach (PCIT) [63] and the regulatory impact factor (RIF) metric were used to identify key transcription factors (KeyTF) from gene expression data [26, 27, 29]. The gene co-expression analysis, performed by Bioconductor package CeTF [26, 27] calculated RIF1, which captures TF showing differential connectivity to DEGs found in contrast between breeds, and RIF2, that focuses on TF showing evidence as predictors of change in abundance of genes with differential expression between breeds (S10 Table). CeTF analysis of LPS treated MDMs displaying all DEGs associated to KeyTF for each bovine breed were plotted in the Cystoscape software and then overlapped by Diffany plugin [34] to highlight genes that were exclusive enriched for biological processed according to each bovine breed (Fig 4). Co-expression networks of Holstein and Gir LPS stimulated MDM revealed various genes detected in genome wide association studies that aim to improve genomic breeding indices (Fig 5 and S11 Table), e.g., milk production [64–68], clinical and subclinical mastitis [69–72], puberty [68, 73–75], feed efficiency [76], adaptation to ecologic conditions [77–79] and cellular and humoral immune responses [49, 80–82]. The biological process enrichment after co-expression analysis also stood out the importance of leukocyte migration and extracellular matrix organization, both controlled by chemokines and cytokines produced by macrophages in inflammation triggered by LPS in Gir and Holstein MDMs. Interestingly, a recent genome wide association study listed the top 10 SNPs that explain 5.05% of *B. bovis* infection level additive genetic variance and identified 42 candidate genes involved in chemokine signalling, extracellular matrix organization and NO production biological mechanisms that might underlie *B. bovis* resistance in cattle [83].

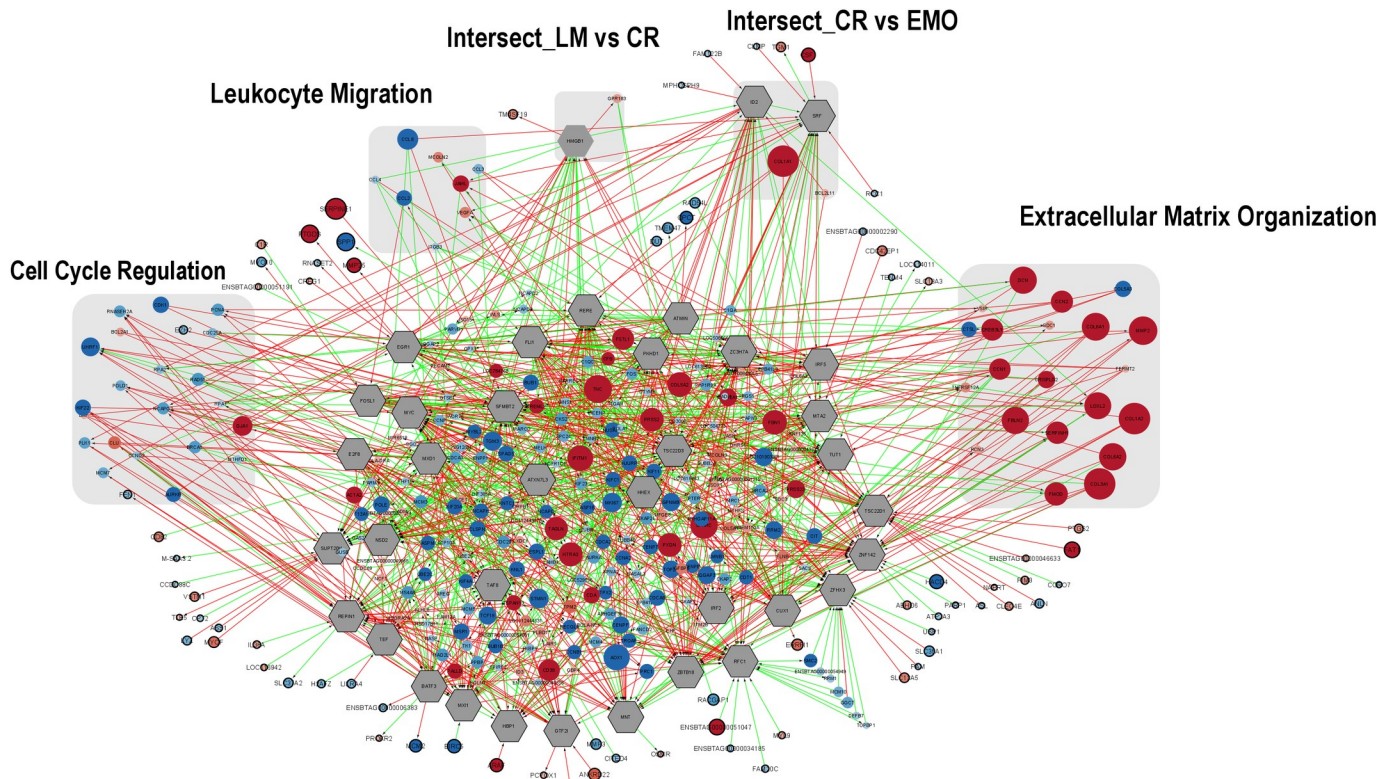

**Fig 4. Meaningful gene-gene associations in co-expression networks of LPS stimulated MDM from Holstein and Gir breeds emphasizing ontology-based differential frameworks enrichment analysis.** Network displaying differential interactions among DEGs of MDMs in Holstein (n = 4) and Gir (n = 4) breeds after co-expression analysis using CeTF and Diffany Cytoscape plugin. Red edges indicate decreased connections in Gir network compared to Holstein LPS stimulated MDMs after 48 hours. Green edges indicate increased connections in Gir network compared to Holstein MDMs in the same condition. Gray hexagonal nodes indicate key transcriptional factors (KeyTF) obtained after PCIT/RIF analysis in CeTF (Bioconductor package). Circular red nodes indicate up-regulated genes while blue nodes indicate downregulated genes found in comparison of MDMs from Holstein and Gir breeds stimulated with LPS and enriched after PCIT/RIF analysis. Lighter and darker hues, as well as the size of the circle, are associated to representativeness of gene expression in the analyses. Genes that displayed unique interaction with KeyTF are highlight by thicker edges. The key ontologies related to Biological Processes differently enriched for Holstein and Gir breeds after co-expression analyses are highlight in grey boxes. LM: Leukocyte Migration; CR: Cell Cycle Regulation; EMO: Extracellular Matrix Organization; Intersect: Genes enriched for more than one process highlighted by co-expression analysis. Genes not related to any Biological process but detected in RIF/PCIT/ Dyffany analyses were grouped in the middle of the figure. All statistical analyses were performed according to software and plugin default parameters and significance thresholds were P<0.001 and FDR<0.05.

## Differential expression of proinflammatory genes in buffy coat cells and LPS treated MDMs from Holstein and Gir cattle and their influence on NO production

Literature findings, especially in murine experimental models represented by C57BL/6 and BALB/c mice, showed that M1 macrophages display a proinflammatory phenotype associated to pathogen-killing abilities while M2 macrophages promote cell proliferation and tissue repair [84]. The macrophage polarization phenotypes have been well characterized in humans and mice influencing on the outcome of infections and parasitic diseases. Unfortunately, M1/M2 macrophage profiling in cattle has been far less explored, mainly for taurine and indicine cattle which are two bovine subspecies that share a common ancestor [85] and display different levels of resistance to infections and parasitic diseases [3, 5, 86]. In order to evaluate if Gir (indicine breed) are committed to a more pro inflammatory status than Holstein animals (taurine breed), the expression of genes associated to M1/M2 phenotypes of inflammatory outcomes such as *Ornithine Aminotransferase (OAT), NRROS, IL-10, TLR4* was evaluated in buffy coat

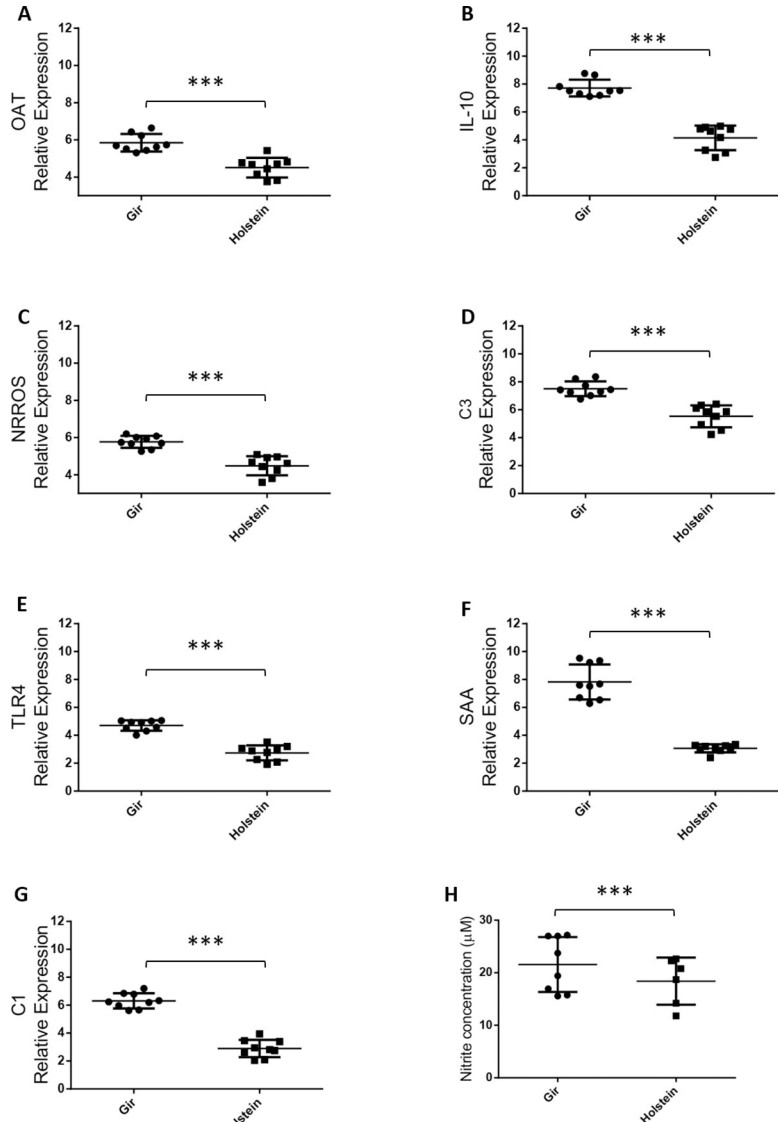

**Fig 5. Differential systemic inflammatory responses in Holstein and Gir cattle.** (A-G) Gene expression of inflammatory molecules from Holstein and Gir buffy coat cells by RT-qPCR. All data were shown as target genes ΔCt average ± SD. *** P<0.001, T-test. (H) Nitrite concentration in serum from Holstein and Gir breeds in homeostatic conditions free of pathogens and drug treatment for three months. Data are shown as concentration average ± SD for each group. T-test was used for comparisons between breeds (*P<0.05).

cells from both breeds. The gene expression of *complement factors 1* and *3* (*C1* and *C3*) and the acute phase protein *SAA* were also evaluated since these molecules are responsible for inflammation amplification [87, 88] and were constantly observed in MDMs transcriptome analysis. The RT-qPCR results from buffy coat cells indicate that all analysed genes were less expressed in Holstein than in Gir samples (Fig 5A–5G; P<0.001). The same genes evaluated in buffy coat cells were also used to validate the MDMs RNA sequencing data (S2A–S2J Fig). The expression of these genes and the ones related to inflammatory signalling pathways such as *NOS2, Interleukin 1 Receptor Associated Kinase 1* (*IRAK1*), *factor nuclear kappa B* (*NFKB2*) matched to the transcriptome analysis, except for IL10 which was not found as DEG at any contrast in RNA sequencing although differently expressed in RT-qPCR (S2A–S2J Fig).

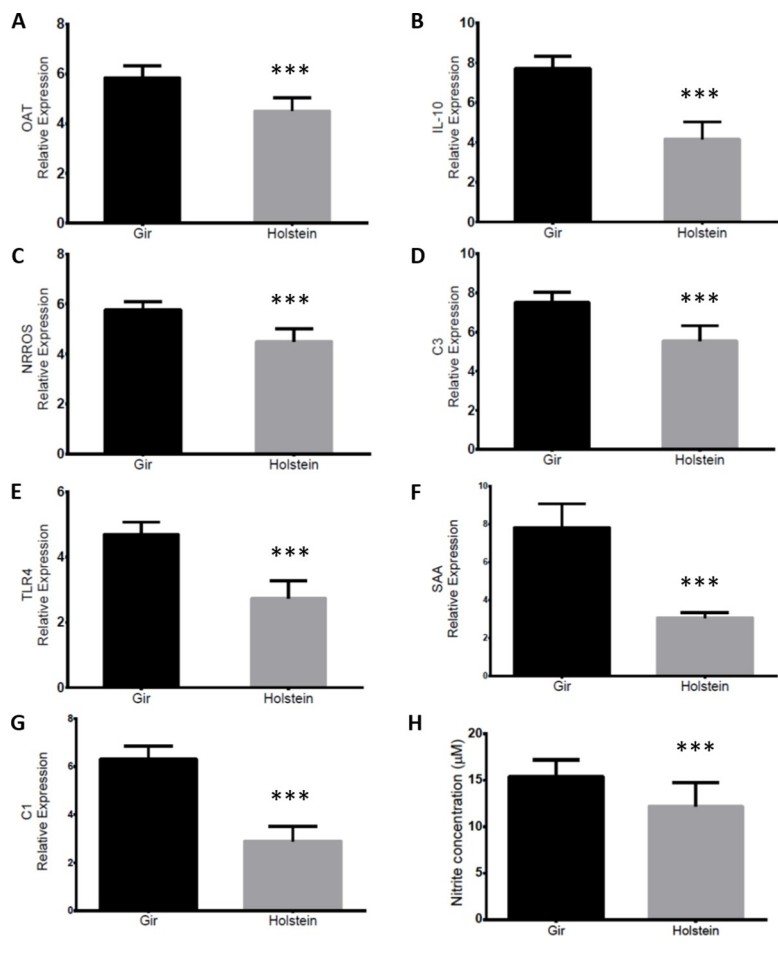

**Fig 6.**

The NOS2 enzyme has an active role in the reactive oxygen species production (ROS) and nitric oxide synthesis pathway induced by LPS [89]. ROS produced by phagocytes are essential for host defense against bacterial and fungal infections [90]. In many cases, resistance of C57BL/6 mice is due to the microbicidal effect of nitric oxide (NO) produced by macrophages in response to interferon-γ (IFN-γ) and tumour necrosis factor-α (TNF-α), mainly secreted by Th1 cells and macrophages, respectively. On the other hand, BALB/c mice are usually partially able to give rise to efficient Th1 lymphocytes and does not control certain infections [18]. Our results indicate that higher expression of *NOS2* gene (S5 Table) was related to the increased NO production in unstimulated MDMs culture supernatant and serum from Gir animals when compared to Holstein (Fig 5H and S2K Fig). It is noteworthy that recently publications indicated that genetic background of the bovine breed affects 77% of phenotypic NO production of MDMs in response to *Escherichia coli* (*E. coli*), *in vitro* [91, 92]. Indeed, NO production and release seems to mediate resistance to many bovine infections, especially for *Mycobacterium bovis*, *Babesia bovis* and *E. coli* in bovine hosts [83, 92, 93]. Conversely, excessive ROS can cause collateral tissue damage during inflammatory processes and therefore is tightly regulated [90]. Gir buffy coat cells also showed an increased gene expression of the *Negative Regulator of ROS* (*NRROS*; Fig 5C), which is the main mechanism that regulates reactive oxygen species production [90]. Gir MDMs also showed higher gene expression of *NOD2* and *FAS* (S5 Table), both related to cellular death process in the presence of larger amounts of NO [94]. In sum, the observed resistance to

various infections in bovine could be associated with this increased ROS production and macrophages microbicidal activity during their homeostatic and inflammatory state, similarly as found in the C57BL/6 murine model. However, there might be differences in response patterns performed by bovine and murine macrophages since M2 markers, such as genes *OAT* and *IL-10*, were also increased in the Gir buffy coat ([Fig 6A and 6B]). Thus, additional specific cellular activation assays should be performed to more accurately understand the development of M1 and M2 response in the bovine breeds and its influence on the outcome of immune response. These might provide insights into the immunological regulation of LPS triggered immune response in cattle, as well as reveal the potential to include immune response traits in genomic selection panels to decrease the occurrence of disease and improve animal health.

## Conclusions

In summary, this study investigated the LPS effect on differential gene expression associated with divergent MDMs phenotypic profiles in Holstein and Gir bovine breeds. Our results showed that these animals differently express genes possibly associated with divergent macrophage polarization. The extracellular matrix organization, leukocyte migration and cell cycle were the most affected biological processes. Differences in macrophage activation between taurine and indicine cattle might be useful to improve animal breeding programs through the use of genomic selection focused to decrease the occurrence of diseases and improve animal health. In addition, our results might help to open new windows to the development of novel technologies to pathogens control as new functional drugs, vaccines and adjuvants based on the bovine genotypic and phenotype profile.

## Supporting information

**S1 Fig. MDMs transcripts mapped data and mRNA representativeness.** Percentage of reads mapped to ARS-UCD1.2 bovine reference genome. (A) Reads mapped for each breed and treatment (unstimulated and LPS); (B) Total number of transcripts detected and categorized according to average of *in vitro* gene expression levels (CPM, counts per million) in unstimulated and LPS treated MDMs from Holstein ang Gir breeds.
(TIF)

**S2 Fig. RNA sequencing validation and nitrite dosage at supernatant of unstimulated and LPS treated MDMs.** (A-J) RT-qPCR of unstimulated and LPS-treated MDMs from Holstein (n = 4) and Gir (n = 4) bovine breeds. Data shown as average ± SD of three replicates for each animal. T-test was used for comparisons between breeds and one-way analysis of variance between different stimuli into the same breed. *P<0.05, **P<0.01, ***P<0.001. (K) Levels of nitrite at unstimulated and LPS (10ng/ml) treated MDM cell culture supernatant from Holstein and Gir breeds after 48hours of stimulation. Data shown as concentration average ± SD for each group. T-test was used for comparisons between breeds (*P<0.05) and one-way analysis variance between different stimuli within the same breed.
(TIF)

**S1 Table. Primer sequences used in RT-qPCR analyses.** Gene symbol, name and primer sequence of all primers designed for RT-qPCR analyses. *RPLP0* and *Ubiquitin* used as reference genes (lowest values of average expression stability M according to GeNORM). Tm: melting temperature; Fwd: forward primer; Rev: reverse primer.
(PDF)

**S2 Table. List of DEGs from unstimulated vs LPS treated Holstein MDMs.** Differential expression was performed on RNA sequencing data from unstimulated and LPS (100ng/ml)

treated MDMs from Holstein breed. Genes that showed statistical differences in contrast (LogFC≥1; CPM>1; FDR<0.05) are shown.
(PDF)

**S3 Table. List of DEGs from unstimulated vs LPS treated Gir MDMs.** Differential expression was performed on RNA sequencing data from unstimulated and LPS (100ng/ml) treated MDMs from Gir breed Genes that showed statistical differences in contrast (LogFC≥1; CPM>1; FDR<0.05) are shown.
(PDF)

**S4 Table. List of DEGs from Holstein vs Gir contrast for unstimulated MDMs.** Differential expression was performed on RNA sequencing data from unstimulated MDMs between Holstein and Gir breeds. genes that showed statistical differences in contrast (LogFC≥1; CPM>1; FDR<0.05) are shown.
(PDF)

**S5 Table. List of DEGs from Holstein vs Gir contrast for LPS treated MDMs.** Differential expression was performed on RNA sequencing data from LPS treated (100 ng/ml) MDMs between Holstein and Gir breeds. Genes that showed statistical differences in contrast (LogFC≥1; CPM>1; FDR<0.05) are shown.
(PDF)

**S6 Table. DEG enrichment analysis of Holstein vs Gir unstimulated MDMs.** DEG enrichment analysis performed by DAVID with data from unstimulated MDMs from Holstein versus Gir animals, showing biological processes and associated genes with statistical significance (P value and FDR). The "Count" column shows the number of enriched genes for each process.
(PDF)

**S7 Table. DEG enrichment analysis of unstimulated vs LPS treated MDMs from Holstein breed.** DEG enrichment analysis performed by DAVID with data from unstimulated versus LPS treated MDMs from Holstein breed, showing biological processes and associated genes with statistical significance (P value and FDR). The "Count" column shows the number of enriched genes for each process.
(PDF)

**S8 Table. DEG enrichment analysis of unstimulated vs LPS treated MDMs from Gir breed.** DEG enrichment analysis performed by DAVID with data from unstimulated versus LPS treated MDMs from Gir breed, showing biological processes and associated genes with statistical significance (P value and FDR). The "Count" column shows the number of enriched genes for each process.
(PDF)

**S9 Table. DEG enrichment analysis of Holstein vs Gir LPS treated MDMs.** DEG enrichment analysis performed by DAVID with data from LPS treated MDMs from Holstein versus Gir animals, showing biological processes and associated genes with statistical significance (P value and FDR). The "Count" column shows the number of enriched genes for each process.
(PDF)

**S10 Table. Bovine key transcription factors (TF) resulting from co-expression analysis of LPS treated MDMs from Gir and Holstein breeds.** Table showing key transcription factors (KeyTF) displaying the scores for RIF1: TF that are consistently most differentially co-expressed with the highly abundant and highly DEGs in Gir and Holstein MDMs; RIF2: TF

with the most altered ability to predict the abundance of DEGs in Gir and Holstein MDMs. The frequencies for each KeyTF were calculated for Holstein and Gir MDM stimulated with LPS. The differential frequencies were also calculated for each KeyTF in order to infer their importance on MDM response to LPS treatment for each Holstein and Gir breeds. (PDF)

**S11 Table. List of DEGs from bovine MDMs that directly interacts to unique key transcription factors which are related to genome wide association studies.** CeTF co-expression analysis of LPS treated MDMs displayed all DEGs associated to KeyTF for each bovine breed. Overlap of co-expression networks in the Cystoscape software with Diffany plugin revealed genes found in genome wide association studies that make one unique connection to KeyTF. (PDF)

## Acknowledgments

We thank our colleagues Klinger de Souza, Geovane Gonçalves de Souza and Michelle de Souza Muniz from Embrapa who provided essential assistance in animal management and sample collection. We also thank the students Ana Flávia Silva Heleno, Mariana Barbosa Pereira and Thiago de Almeida Oliveira for laboratory assistance during the cell culture assays.

## Author Contributions

**Conceptualization:** Marco Antônio Machado, Wanessa Araújo Carvalho.

**Data curation:** Raquel Morais de Paiva Daibert, Carlos Alberto Oliveira de Biagi Junior, Marco Antônio Machado, Wanessa Araújo Carvalho.

**Formal analysis:** Raquel Morais de Paiva Daibert, Carlos Alberto Oliveira de Biagi Junior, Felipe de Oliveira Vieira, Marcos Vinicius Gualberto Barbosa da Silva, Hyago Passe Pereira, Marco Antônio Machado, Wanessa Araújo Carvalho.

**Funding acquisition:** Marco Antônio Machado, Wanessa Araújo Carvalho.

**Investigation:** Raquel Morais de Paiva Daibert, Carlos Alberto Oliveira de Biagi Junior, Felipe de Oliveira Vieira, Marcos Vinicius Gualberto Barbosa da Silva, Eugenio Damaceno Hottz, Mariana Brandi Mendonça Pinheiro, Daniele Ribeiro de Lima Reis Faza, Marta Fonseca Martins, Humberto de Mello Brandão, Wanessa Araújo Carvalho.

**Methodology:** Marcos Vinicius Gualberto Barbosa da Silva, Eugenio Damaceno Hottz, Mariana Brandi Mendonça Pinheiro, Daniele Ribeiro de Lima Reis Faza, Marco Antônio Machado, Wanessa Araújo Carvalho.

**Project administration:** Marco Antônio Machado, Wanessa Araújo Carvalho.

**Resources:** Marta Fonseca Martins, Humberto de Mello Brandão, Marco Antônio Machado, Wanessa Araújo Carvalho.

**Supervision:** Marco Antônio Machado, Wanessa Araújo Carvalho.

**Validation:** Raquel Morais de Paiva Daibert, Carlos Alberto Oliveira de Biagi Junior, Wanessa Araújo Carvalho.

**Visualization:** Raquel Morais de Paiva Daibert, Eugenio Damaceno Hottz, Hyago Passe Pereira, Marco Antônio Machado, Wanessa Araújo Carvalho.

**Writing – original draft:** Raquel Morais de Paiva Daibert, Carlos Alberto Oliveira de Biagi Junior, Wanessa Araújo Carvalho.

**Writing – review & editing:** Eugenio Damaceno Hottz, Marta Fonseca Martins, Marco Antônio Machado, Wanessa Araújo Carvalho.

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
