## [Decision Letter · Decision Letter 0]

26 May 2020

PONE-D-20-11705

Lipopolysaccharide triggers different transcriptional signatures in taurine and indicine cattle macrophages: reactive oxygen species and potential outcomes to the development of immune response to infections

PLOS ONE

Dear Dr. Wanessa Araujo Carvalho,

Thank you for submitting your manuscript to PLOS ONE. After careful consideration, we feel that it has merit but does not fully meet PLOS ONE’s publication criteria as it currently stands. Therefore, we invite you to submit a revised version of the manuscript that addresses the points raised during the review process.

I asked 2 external reviewers to examine your submission and their comments are attached and their recommendations are somewhere between minor and major recommendations. One reviewer was an expert bioinformatician, while the other a specialist in bovine infections, so they raise different points that have to be addressed. 

We look forward to receiving your revised manuscript.

Kind regards,

Gordon Langsley

Academic Editor

PLOS ONE

Journal Requirements:

2. In your Methods section, please state the volume of the blood samples collected for use in your study.

3. In your Methods section, please include a comment about the state of the animals following this research. Were they euthanized or housed for use in further research? If any animals were sacrificed by the authors, please include the method of euthanasia and describe any efforts that were undertaken to reduce animal suffering.

4. We note that you are reporting an analysis of a microarray, next-generation sequencing, or deep sequencing data set. PLOS requires that authors comply with field-specific standards for preparation, recording, and deposition of data in repositories appropriate to their field. Please upload these data to a stable, public repository (such as ArrayExpress, Gene Expression Omnibus (GEO), DNA Data Bank of Japan (DDBJ), NCBI GenBank, NCBI Sequence Read Archive, or EMBL Nucleotide Sequence Database (ENA)). In your revised cover letter, please provide the relevant accession numbers that may be used to access these data. For a full list of recommended repositories, see http://journals.plos.org/plosone/s/data-availability#loc-omics or http://journals.plos.org/plosone/s/data-availability#loc-sequencing.

Additional Editor Comments (if provided):

Reviewers' comments:

Reviewer's Responses to Questions

**Comments to the Author**

1. Is the manuscript technically sound, and do the data support the conclusions?

Reviewer #1: Partly

Reviewer #2: Yes

2. Has the statistical analysis been performed appropriately and rigorously? 

Reviewer #1: Yes

Reviewer #2: Yes

3. Have the authors made all data underlying the findings in their manuscript fully available?

Reviewer #1: No

Reviewer #2: No

4. Is the manuscript presented in an intelligible fashion and written in standard English?

Reviewer #1: No

Reviewer #2: Yes

5. Review Comments to the Author

Reviewer #1: From indicine and taurine

bovine blood samples, the authors induce macrophages from monocytes. The macrophages are then treated with (and without) lipopolysaccharide, followed by RNA-Seq. Using conventional software, differential genes are detected and compared between species and between treated and untreated samples. Further, RT-qPCR is performed for a selection of target genes.

From this, the authors find distinct differences in the immune responses between indicine and taurine. The topic is timely and interesting findings are presented.

I find it difficult to understand exactly how the RNA sequencing results and the qPCR validations relate to each other. Figure 6 shows a number of genes not appearing in Figure 3, suggesting to me that the authors are validating a set of genes based on their a priori knowledge, rather than confirming the results obtained from the RNA-Seq? This may of course not be a problem in itself as long as the authors clearly state and justify this.

More importantly, assuming that the genes from Figure 6 are not hidden somewhere in Figure 3 (and just not mentioned by name), there should be a thorough comparison between the RNA-Seq and the qPCR results. Calling DGE genes is a fairly arbitrary process depending on the specific threshold levels applied, so it would be naive to expect an exact correspondence between DGE genes and qPCR results. But it is essential to know if these different experimental approaches are in direct contrast to each other. Besides this, the expression levels, fold-change and FDR values should be presented as supplementary material.

In fairness, the authors state (line 465) that some findings are from "different experimental strategies". But this nevertheless gives the reader a sense of cherry picking results that fit conventional wisdom within the field.

Figure 6: It is unclear to me why these genes are suddenly tested in duffy blood? Why not use the MDM results from FigS2? (line 374 in the main text refers to FigS2 as "RNA sequencing" but it appears to be RT-qPCR results as well, which is also stated in its legend). And why are the same genes not tested (or presented) for both duffy and MDM?

Figure 3: It is not clear to me what the Z-scores reflect. Are they calculated between gir and holstein samples, or within each sample? Intuitively I would expect that green/positive scores meant up-regulated, but on line 508 it is stated that NOS2, C3, IL36β and CCL24 are generally up-regulated in gir. Yet, only CCL24 appears to be 'green' in gir (so behaving opposite of the 3 other genes in any case - does "Most of the DEGs for inflammatory response biological process" then mean 3 out of 4?). A clearer explanation of this figure - and the specific procedure behind it - should be provided.

To my knowledge, the RNA input material does exceed the recommendations for the TruSeq kit used in the study, yet it would be comforting to know the level of sequence redundancy between reads, ensuring that read abundance is not affected by biases during the library build. The samtools and picard software suites have tools to mark duplicate reads.

Table 1 shows that the similar sequencing efforts between samples result in roughly even numbers of detected transcripts. Importantly, this does not reveal if the samples are exhausted, i.e. if deeper sequencing would result in a significantly higher number of detected transcripts. To test if a plateau of detected transcripts is reached would require a downsizing of the read data and subsequent re-analysis of such subsets. As this study aims to characterize major/general differences between the species, such a test is most likely not essential in this case. Yet, it would suit the manuscript of the authors briefly acknowledged this issue.

lines 330-2: "It is also worth noting that activation of neutrophil

degranulation pathways was detected in both breeds (S2 and S3 Tables), however

showing different significances (taurine P=6.69E-04; indicine P= 0.01) suggesting a

differential role of these cells in the biological responses of these breeds."

Without knowing exactly how these p-values are produced (they are likely dependent on values derived from the entire pool of genes and not just the subset of genes being tested), these relatively subtle differences cannot form the basis for such speculations on different biological.

The manuscript will need a language revision. On the first page of the introduction I found the following examples:

"which reflects in variable response"

"related with"

"stimulates 40 host cells as monocytes/macrophages"

"biding"

"not capable to reproduce"

Reviewer #2: The study of Carvalho et al., investigates the differential responses of Hostein (bos Taurus) and Gir (bos indicus) monocytes derived macrophages (MDMs) stimulated with LPS. The authors used RNAseq approaches, all procedures and step of analysis are well documented. The results showed a breed difference, suggesting a possibly divergent macrophages polarization (like the M1/M2 balance, well described in human and mice). I found the results sound and worthy of publication. I have the following specific comments/questions for the authors.

Major:

_ 4 biological replicates per group were used for the RNAseq. Why not n=6/group like for some other analysis in the manuscript?

_ The UMD3.1.S4 genome was published in 2014. A new assembly version is available since 2018, under the name ARS-UCD1.2. Why not using the most recent version?

_ The foldChange (FC) values considered by the authors were equal or superior to 1. This is not very stringent. Publication commonly used FC cut-off >1.5 or 2. Is the analysis still robust with a higher cut-off?

_ The numbers of biological replicates (n=X) is lacking in the legend of figures 4, 5, 6 and S2.

_ line 195 “RPLO and Ubiquitin were selected as housekeeping genes”. To meet the MIQE guidelines and publish qPCR data, authors should use at least 3 housekeeping genes for all experiments, even though then have tested 4 or 5 before selecting 2. Please include another housekeeping gene.

_ Can the authors comment on the inter-individual variability among Holstein and Gir. Is it equivalent? Is it possible to show individual data, and median +/- interquartile range on the figures.(FigS2 for instance)

_ Figure 3: the color code Green/Red is not visible by color blind people. Blue/orange or Blue/red are common colorblind-friendly palette.

_ Figure 5 is interesting but gene names are not readable. Can the authors add a table with the 70 genes which showed an unique direct interaction to key transcription factors, and highlight the 22 which are associated to mastitis resistance.

_ Lines 330-332 and 494-496: The authors have seeing the neutrophil degranulation pathways in both breeds, with different significances. They concluded that neutrophil could play a differential role in Holstein and Gir. I do not agree with this conclusion. If the granuloctes pathway is significantly found in the two groups, a higher p-value in one group does not mean a higher implication of this pathway. The authors suggest a stronger role for neutrophils in Holstein (lines 494-495), but in the S2 and S3 Tables, more genes are present in the Gir RNAseq data (19 genes in the data set /480 genes in the pathway) than in Holstein’s (11/480).

Minor:

_ Data are not fully available because the authors are using them to work on another project and have to respect intellectual property restrictions. For the reviewing process, RNA sequence data are supposed to be available from GEO repository, but I could not find the number 147813 on the GEO website.

https://www.ncbi.nlm.nih.gov/gds/?term=147813

_ The authors used alternatively three different names for the animals: bos Taurus/bos indicus; or taurine/indicine; or Holstein/Gir. I would suggest to use the same terminology in the manuscript. For instance the first time Holstein (Bos Taurus) and Gir (bos indicus); and then “Holstein” and “Gir” in the rest of the manuscript and for the figures. I have the feeling that taurine and indicine are less obvious for the general audience but I may be wrong. My point is please use the same terminology everywhere in the article.

_ Line 195 “4 housekeeping genes”, but 5 are listed into brackets. Please correct.

_ Quality of figure 2 should be improved (too many pixels).

_ Figure 5: “ECM” is not defined in the legend.

_ Figure S1: what is “MNP1” in orange in the CD14 histogram?

6. PLOS authors have the option to publish the peer review history of their article (what does this mean?). If published, this will include your full peer review and any attached files.

Reviewer #1: Yes: Tobias Mourier

Reviewer #2: Yes: Aude Remot

---

## [Author Response · Author response to Decision Letter 0]

23 Sep 2020

Reviewers’ Comments and Authors Response

Paper number: PONE-D-20-11705

Paper title: Lipopolysaccharide triggers different transcriptional signatures in taurine and indicine cattle macrophages: reactive oxygen species and potential outcomes to the development of immune response to infections

Authors: Raquel M. P. Daibert, Carlos A. O. Biagi Junior, Felipe O. Vieira, Marcos V.G.B. Silva, Eugenio D. Hottz, Mariana B.M. Pinheiro, Daniele R.L.R. Faza, Hyago P. Pereira, Marta F. Martins, Humberto M. Brandão, Marco A. Machado, Wanessa A. Carvalho

The authors would like to thank the area editor and the reviewers for their precious time and invaluable comments. We have carefully addressed all the comments. The corresponding changes and refinements made in the revised manuscript are summarized below. 

EDITOR’S COMMENTS AND QUESTIONS 

Author: We prepared a better revision to meet Plos One style requirement according to the suggestions above. Some modifications in the manuscript text were made in order to make it clearer and easier for the readers to understand, including merging the results and discussion sections. 

2. In your Methods section, please state the volume of the blood samples collected for use in your study.

Author: The volume of blood samples collected to be used in our study was 60ml per animal. This information was added to the Methods section (Lines 83-4). 

3. In your Methods section, please include a comment about the state of the animals following this research. Were they euthanized or housed for use in further research? If any animals were sacrificed by the authors, please include the method of euthanasia and describe any efforts that were undertaken to reduce animal suffering.

Author: All animals were housed to be used in additional research after our experimental trial finished. This statement was included in the Methods section “Animals”. 

4. We note that you are reporting an analysis of a microarray, next-generation sequencing, or deep sequencing data set. PLOS requires that authors comply with field-specific standards for preparation, recording, and deposition of data in repositories appropriate to their field. Please upload these data to a stable, public repository (such as ArrayExpress, Gene Expression Omnibus (GEO), DNA Data Bank of Japan (DDBJ), NCBI GenBank, NCBI Sequence Read Archive, or EMBL Nucleotide Sequence Database (ENA)). In your revised cover letter, please provide the relevant accession numbers that may be used to access these data. For a full list of recommended repositories, see http://journals.plos.org/plosone/s/data-availability#loc-omics or http://journals.plos.org/plosone/s/data-availability#loc-sequencing.

Author: We apologize for our misunderstanding in providing the accession number and token for GEO repository only at the manuscript submission system, but not in the last cover letter. Actually, after revision of the manuscript we changed the access policy for all RNA sequence datafiles, which are now available from the GEO repository database (https://www.ncbi.nlm.nih.gov/geo; accession number GSE 147813) open for public access.

REVIEWERS' COMMENTS AND QUESTIONS

Reviewer #1: From indicine and taurine bovine blood samples, the authors induce macrophages from monocytes. The macrophages are then treated with (and without) lipopolysaccharide, followed by RNA-Seq. Using conventional software, differential genes are detected and compared between species and between treated and untreated samples. Further, RT-qPCR is performed for a selection of target genes. From this, the authors find distinct differences in the immune responses between indicine and taurine. The topic is timely and interesting findings are presented.

1.1. I find it difficult to understand exactly how the RNA sequencing results and the qPCR validations relate to each other. Figure 6 shows a number of genes not appearing in Figure 3, suggesting to me that the authors are validating a set of genes based on their a priori knowledge, rather than confirming the results obtained from the RNA-Seq? This may of course not be a problem in itself as long as the authors clearly state and justify this.

More importantly, assuming that the genes from Figure 6 are not hidden somewhere in Figure 3 (and just not mentioned by name), there should be a thorough comparison between the RNA-Seq and the qPCR results. Calling DGE genes is a fairly arbitrary process depending on the specific threshold levels applied, so it would be naive to expect an exact correspondence between DGE genes and qPCR results. But it is essential to know if these different experimental approaches are in direct contrast to each other. Besides this, the expression levels, fold-change and FDR values should be presented as supplementary material.

AUTHOR: All the previous data were reanalysed using the newest bovine genome assembly (ARS-UCD1.2 submitted by USDA/ARS on April 2018) as suggested by other reviewer. The new alignment resulted in a higher number of DEGs in all contrasts although the enrichment continues to point to the same major biological processes. The global conclusion of the article did not alter, suggesting that Gir (indicine cattle) shows a natural propensity to generate a M1 profile inflammatory immune response displaying increased activation efficiency of antigen presentation pathways, oxygen reactive production and leukocyte activation and recruitment. Some modifications in the manuscript text were made in order to make it clearer and easier for the readers to understand, including merging the results and discussion sections. We also added a short explanation of how we have chosen target genes for validation of RNA sequencing and evaluation of buffy coat inflammatory responses. Briefly, the genes used in RT-qPCR to validate RNA sequencing results were chosen after enrichment analysis of DGE found between LPS treated and untreated monocytes differentiated to macrophages (MDMs) from Holstein and Gir animals. Since “immune response” and “inflammatory immune response” were both enriched in DGE found between taurine and indicine MDMs (S6 and S9 Tables), genes were selected when they exhibited a key role in the differentiation of macrophage phenotype (M1 and M2), in cell activation status and LPS triggered innate immune response. For buffy coat cells we selected genes related to complement and acute phase proteins since they mediate inflammation development and were also produced by monocytes and macrophages. They were highlighted in the co-expression analysis (Fig 4 and S11 Table). 

Regarding the gene expression levels, fold-change and FDR values from RNA sequencing analysis, a complementary material in this new version of the manuscript was supplied (S2-5 Tables). Indeed, RNA sequence data is also available from the GEO repository (https://www.ncbi.nlm.nih.gov/geo, accession number GSE147813).

1.2. In fairness, the authors state (line 465) that some findings are from "different experimental strategies". But this nevertheless gives the reader a sense of cherry picking results that fit conventional wisdom within the field.

AUTHOR: The term “different experimental strategies” was removed. We revised the manuscript text and hope that this kind of written did not happen in this new version. 

1.3. Figure 6: It is unclear to me why these genes are suddenly tested in buffy blood? Why not use the MDM results from FigS2? (line 374 in the main text refers to FigS2 as "RNA sequencing" but it appears to be RT-qPCR results as well, which is also stated in its legend). And why are the same genes not tested (or presented) for both buffy and MDM?

AUTHOR: We tested the buffy coat cells in order to establish an association of in vitro MDM transcriptomes to the peripheral immune response which were preferentially displayed by Holstein and Gir breeds under homeostatic conditions. To validate RNA sequencing data, we tested genes that are considered classical markers for M1 and M2 macrophages phenotype (NOS2, NRROS, IL10, OAT) and important for MDM activation of pro inflammatory status (TLR4, IRAK2, NFKB2, GATA-3, BMPR1, ENGL3, C3). Since our RNA sequencing results pointed to a more prominent M1 macrophage phenotype in the Gir (indicine breed), we investigated if it might be involved in differences that mediate development of divergent patterns of systemic immune response. By using buffy coat cells to access the levels of transcripts for molecules that are involved with inflammatory and innate immune responses, we observed that Holstein (taurine breed) displayed lower expression of all selected genes, especially the ones involved in production of oxygen reactive species and complement. This emphasized the hypothesis that taurine animals might have a compromised innate immune response that might influence T helper development responses when compared to indicine animals. Nevertheless, our team realize that additional experiments must be done in order to validate this information. Various literature references were cited in the manuscript discussion in order to support this information. We also did a nitrite dosage in the serum of all animals involved in the experiment and showed that taurine animals showed lower levels of this molecule which might implicate in impairment of Th1 responses and higher susceptibility to infection and parasite diseases, also discussed along the manuscript. So, we did not test the same genes for macrophage and buffy coat since we were trying to answer different questions that corroborate the unique hypothesis that Holstein and Gir bovine breeds may show different immune responses due to the macrophage transcriptional signatures. We hope that this new version of the manuscript is better written in order to promote a clearer interpretation of our hypothesis and results.

1.4. Figure 3: It is not clear to me what the Z-scores reflect. Are they calculated between gir and holstein samples, or within each sample? Intuitively I would expect that green/positive scores meant up-regulated, but on line 508 it is stated that NOS2, C3, IL36β and CCL24 are generally up-regulated in gir. Yet, only CCL24 appears to be 'green' in gir (so behaving opposite of the 3 other genes in any case - does "Most of the DEGs for inflammatory response biological process" then mean 3 out of 4?). A clearer explanation of this figure - and the specific procedure behind it - should be provided.

Author: We did apply changes in colour and realized that we forgot to cluster the columns (sample information) in the Figure 3. That’s why it was behaving as the opposite of mentioned in the previous manuscript. In the heatmap, Z-scores were calculated for each row (each gene) and each column (each sample) and plotted accordingly to the normalized expression values. This consists in subtracting the mean and dividing by the standard deviation, which is a step after clustering the data that only affects the graphical look and improves the colour visualization. 

1.5. To my knowledge, the RNA input material does exceed the recommendations for the TruSeq kit used in the study, yet it would be comforting to know the level of sequence redundancy between reads, ensuring that read abundance is not affected by biases during the library build. The samtools and picard software suites have tools to mark duplicate reads.

Author: We are sorry for not clearly showed these results in the older version of the manuscript. Now, we carefully described all the data regarding the RNA sequencing and data analysis both at “Materials and methods” (Lines 128-147) and “Results and discussion” (Lines 258-272) sections. Indeed, the duplicated sequences was evaluated in FastQC software, in which Picard tool are used as a plugin to calculate Sequence Duplication Levels (https://www.bioinformatics.babraham.ac.uk/projects/fastqc/). This information is showed in Table 1. The software STAR (Gingeras, T.R. & Dobin, A. Mapping RNA-seq Reads with STAR, Curr Protoc Bioinformatics 51: 11.14.1–11.14.19, 2015, doi: 10.1002/0471250953.bi1114s51) was used to eliminate bias resulted from the alignment of more complex RNA sequence arrangements, such as chimeric and circular RNA. STAR software also aligns spliced sequences of any length with moderate error rates providing scalability for emerging sequencing technologies. If is there any issue that we have not noticed in these analyses, please, let us know. 

1.6. Table 1 shows that the similar sequencing efforts between samples result in roughly even numbers of detected transcripts. Importantly, this does not reveal if the samples are exhausted, i.e. if deeper sequencing would result in a significantly higher number of detected transcripts. To test if a plateau of detected transcripts is reached would require a downsizing of the read data and subsequent re-analysis of such subsets. As this study aims to characterize major/general differences between the species, such a test is most likely not essential in this case. Yet, it would suit the manuscript of the authors briefly acknowledged this issue.

Author: We appreciated your comment and addressed this issue at Lines 267-68 of the corrected manuscript. Indeed, the software HT seq count and STAR, both used to analyse RNA sequences, ignore reads that are common between contrasts. Anyway, thanks a lot for the comment. We will use the suggested strategy for future analyses using deeper sequencing data. 

1.7. Lines 330-2: "It is also worth noting that activation of neutrophil

degranulation pathways was detected in both breeds (S2 and S3 Tables), however showing different significances (taurine P=6.69E-04; indicine P= 0.01) suggesting a differential role of these cells in the biological responses of these breeds." Without knowing exactly how these p-values are produced (they are likely dependent on values derived from the entire pool of genes and not just the subset of genes being tested), these relatively subtle differences cannot form the basis for such speculations on different biological.

AUTHOR: We agreed and deleted the entire sentence.

1.8. The manuscript will need a language revision. On the first page of the introduction I found the following examples:

"which reflects in variable response"

"related with"

"stimulates 40 host cells as monocytes/macrophages"

"biding"

"not capable to reproduce"

AUTHOR: Some language expressions are common for Portuguese speakers but are not accepted for English version. We apologize for grammatical errors and we have made a comprehensive revision in this new version of the manuscript. Thanks a lot for pointing out this issue showing examples. 

Reviewer #2: The study of Carvalho et al., investigates the differential responses of Holstein (Bos taurus) and Gir (Bos indicus) monocytes derived macrophages (MDMs) stimulated with LPS. The authors used RNAseq approaches, all procedures and step of analysis are well documented. The results showed a breed difference, suggesting a possibly divergent macrophages polarization (like the M1/M2 balance, well described in human and mice). I found the results sound and worthy of publication. I have the following specific comments/questions for the authors.

Major comments and questions:

2.1. 4 biological replicates per group were used for the RNAseq. Why not n=6/group like for some other analysis in the manuscript?

AUTHOR: We had problems in RNA sample quality that did not reach the minimal parameters required for sequencing such as minimum of 100ng total RNA and RNA integrity number (RIN) higher than 7.00 as suggested by Agilent and literature findings. For a clearer understanding of that, we altered the description of samples sequenced in the Methods section (lines 128- 138). 

2.2. The UMD3.1.S4 genome was published in 2014. A new assembly version is available since 2018, under the name ARS-UCD1.2. Why not using the most recent version?

AUTHOR: We really appreciated the suggestion and apologize for using an older genome. With your suggestion, we made improvements in our script analysis, which is now updated. We also did a comparative analysis of Ensembl annotation and found a greater number of DEGs under the newer bovine genome assembly (ARS-UCD1.2). Besides that, the enrichment analysis of DEGs indicated similar biological processes highlighted in the former version of the manuscript such as immune response, cell adhesion and division. The immune response processes enriched by DAVID software also displayed high correspondence between DEGs found in UMD3.1.S4 and ARS-UCD1.2 genome which had no impact on the manuscript conclusions after this new alignment. 

2.3. The foldChange (FC) values considered by the authors were equal or superior to 1. This is not very stringent. Publication commonly used FC cut-off >1.5 or 2. Is the analysis still robust with a higher cut-off?

AUTHOR: We apologize for this issue and corrected this information throughout the manuscript text. The threshold values of logFC>1 or <1 and FDR < 0,5 were used to consider a gene upregulated or downregulated, which means a cut off >2. 

2.4. The numbers of biological replicates (n=X) is lacking in the legend of figures 4, 5, 6 and S2.

AUTHOR: Thanks for noticing that. The legends are now corrected.

2.5. Line 195 “RPLO and Ubiquitin were selected as housekeeping genes”. To meet the MIQE guidelines and publish qPCR data, authors should use at least 3 housekeeping genes for all experiments, even though then have tested 4 or 5 before selecting 2. Please include another housekeeping gene.

AUTHOR: Literature findings did not show any deep study for reference genes in bovine animals, especially in MDMs cells. So, we selected four most cited reference genes found on Pubmed database that used qPCR data to support biological information in bovine. GENorm algorithm calculated gene-stability classified according to M values for RPLP0, Ubiquitin, GAPDH and 18 S ribosomal RNA (Vandesompele, J. et al. 2002. Accurate normalization of real-time quantitative RT-PCR data by geometric averaging of multiple internal control genes. Genome Biol 3, research0034.1. https://doi.org/10.1186/gb-2002-3-7-research0034). RPLP0 (M= 0.4) and Ubiquitin (M=0.33) were the two most stable genes based on M values (S1 Table). The Glyceraldehyde 3-phosphate dehydrogenase oxidase (GAPDH) may not be a good choice to be used as a third reference gene because it has been implicated in several non-metabolic processes, including initiation of apoptosis and endoplasmic reticulum to Golgi vesicle shuttling, both triggered by LPS recognition by macrophages (Tarze A. et al. 2007. GAPDH, a novel regulator of the pro-apoptotic mitochondrial membrane permeabilization. Oncogene. 26 (18): 2606–20. doi:10.1038/sj.onc.1210074). The 18 S ribosomal RNA gene showed the highest M value (0.47), which is not recommended for the qPCR analysis in our experiments. These findings added to the fact that we had limited cDNA amount to test additional reference genes made us decide to use the two most stable genes for RT-qPCR analysis. The optimal number and choice of reference genes were experimentally determined accordingly to recommendations of MIQE guidelines (Stephen A Bustin et al. The MIQE Guidelines: Minimum Information for Publication of Quantitative Real-Time PCR Experiments, Clinical Chemistry, Volume 55, Issue 4, 1 April 2009, Pages 611–622, https://doi.org/10.1373/clinchem.2008.112797). It is relevant to mention that all contrasts involving comparison between MDMs transcripts from taurine and indicine breeds in RNA sequencing data showed gene expression match evaluated by RTqPCR. Indeed, the nitric oxide (NO) production detected by quantifying the NO breakdown of the final product nitrite also corroborate the NOS2 and NRROS supposed enzyme activity evaluated by gene expression results from both RNA sequencing and qPCR of MDMs Holstein and Gir cattle samples (Figs6 and S2; S2-5 Tables). In this case, we understand that it might be acceptable the use of two reference genes for qRTPCR assays in our research since their choice was experimentally determined and the methodology was better described in this new version of the manuscript (Lines 194-210). 

2.6.Can the authors comment on the inter-individual variability among Holstein and Gir. Is it equivalent? Is it possible to show individual data, and median +/- interquartile range on the figures.(FigS2 for instance)

AUTHOR: The Gir and Holstein are genetically improved breeds for milk production and show trait consistence required by the breeders associations in Brazil. Dairy cattle industry also uses extensive reproductive and genomic strategies which helps accelerating the genetic progress of the herds. The animals chosen for our research were selected from herds participating in the Holstein and Gir breeding programs coordinated by Embrapa Dairy Cattle. To make it clearer, we have changed all RT-qPCR Figures 5 and S2 in order to show individual data and median +/- interquartile range in each breed. 

 2.7. Figure 3: the color code Green/Red is not visible by color blind people. Blue/orange or Blue/red are common colorblind-friendly palette.

AUTHOR: Thanks a lot for the suggestion. The colours were changed to blue and yellow, also differentiated by colour-blind readers. Unfortunately, in the Figure 5, the green/red edges were not changed because they were generated by the Diffany plugin default and therefore could not be altered by the user. But we understand that this will not affect the interpretation results from colour-blind readers. 

2.8. Figure 5 is interesting but gene names are not readable. Can the authors add a table with the 70 genes which showed an unique direct interaction to key transcription factors, and highlight the 22 which are associated to mastitis resistance.

AUTHOR: In this new version of the manuscript, after the alignment done using the ARS-UCD1.2 newest bovine genome assembly, the numbers of genes that showed an unique direct interaction to key transcription factors increased to 82. We have made a new manual annotation of these genes and highlighted 22 that have been associated to milk production and quality, mastitis susceptibility, fertility, adaptation to ecologic conditions besides of cellular and humoral immune response in whole genome association studies (S11 Table). The macrophages are plastic cells that control not only the inflammatory immune response development but also metabolic pathways associated to lipid and carbohydrate metabolism (Reviewed at Jan Van den Bossche et al. 2017. Macrophage Immunometabolism: Where Are We (Going)? Trends Immunol Jun;38(6):395-406. doi: 10.1016/j.it.2017.03.001 and Murphy, M.P. 2019. Rerouting metabolism to activate macrophages. Nat Immunol 20, 1097–1099. https://doi.org/10.1038/s41590-019-0455-5). Thus, we used genome wide association studies found in the literature to infer about traits related to the macrophage activity in bovine animals and its influence on the outcome of inflammatory and metabolic responses in cattle. In this new version of the manuscript we have also considered in vitro studies with fewer animals, but now we refined the analysis using population studies, which might account for a more accurate interpretation. 

2.9. Lines 330-332 and 494-496: The authors have seeing the neutrophil degranulation pathways in both breeds, with different significances. They concluded that neutrophil could play a differential role in Holstein and Gir. I do not agree with this conclusion. If the granuloctes pathway is significantly found in the two groups, a higher p-value in one group does not mean a higher implication of this pathway. The authors suggest a stronger role for neutrophils in Holstein (lines 494-495), but in the S2 and S3 Tables, more genes are present in the Gir RNAseq data (19 genes in the data set /480 genes in the pathway) than in Holstein’s (11/480).

AUTHOR: We agreed and deleted the sentences related to such strong inferences.

Minor comments and questions:

2.10. Data are not fully available because the authors are using them to work on another project and have to respect intellectual property restrictions. For the reviewing process, RNA sequence data are supposed to be available from GEO repository, but I could not find the number 147813 on the GEO website.

https://www.ncbi.nlm.nih.gov/gds/?term=147813

Author: We apologize for our misunderstanding in providing the accession number and token for GEO repository only at the manuscript submission system, but not in the cover letter. Actually, after revision of the manuscript we changed the access policy for all RNA sequence datafiles, which are now available from the GEO repository database (https://www.ncbi.nlm.nih.gov/geo; accession number GSE 147813) open for public access.

2.11. The authors used alternatively three different names for the animals: bos Taurus/bos indicus; or taurine/indicine; or Holstein/Gir. I would suggest to use the same terminology in the manuscript. For instance the first time Holstein (Bos Taurus) and Gir (bos indicus); and then “Holstein” and “Gir” in the rest of the manuscript and for the figures. I have the feeling that taurine and indicine are less obvious for the general audience but I may be wrong. My point is please use the same terminology everywhere in the article.

AUTHOR: We have standardized this terminology as suggested. 

2.12. Line 195 “4 housekeeping genes”, but 5 are listed into brackets. Please correct.

AUTHOR: We corrected this error and deleted NADPH that was not used in our experiment. 

2.13. Quality of figure 2 should be improved (too many pixels).

AUTHOR: We performed the quality improvement of all figures of the manuscript according PACE software, as indicated by Plos One submission platform. 

2.14. Figure 5: “ECM” is not defined in the legend.

AUTHOR: “ECM” was used as abbreviation of “Extracellular Matrix”. In this new version of the manuscript, after the alignment of our RNA sequences to the newest bovine genome assembly, the biological process related to “Extracellular Matrix” is still highlighted in co-expression analyses. We have corrected the legend of the figure and use “EMO” as abbreviation of the biological process “Extracellular Matrix Organization”. 

2.15. Figure S1: what is “MNP1” in orange in the CD14 histogram?

AUTHOR: This figure was renamed as Fig 1 in order to make the data clearer to readers. The first step in flow cytometry analysis is often make gates that are used to distinguish populations of cells based on their forward and side scatter properties. Forward and side scatter give an estimation of the size and granularity of the cells respectively, although this can depend on several factors of the sample. Distinguishing populations of cells can be relatively straight forward when there is only one type of cell, but it can be more complex for samples where there are multiple cell types. “NMP1” is an abbreviation of “No Marked Population 1” used to separate mononuclear cells from debris and other possible cell contaminants on day 2 of analysis. Since the Figure was chosen as a representation of the flow cytometry assays, we changed the image 1B using the same terminology and gates for both histograms. The bar chart (Fig1A) represents the calculation of % FITC positive cell in all events counted in the gate of mononuclear cells. 

Do you want your identity to be public for this peer review? For information about this choice, including consent withdrawal, please see our Privacy Policy.

Reviewer #1: Yes: Tobias Mourier

Reviewer #2: Yes: Aude Remot

Corresponding author: Yes: Wanessa A. Carvalho

---

## [Decision Letter · Decision Letter 1]

22 Oct 2020

Lipopolysaccharide triggers different transcriptional signatures in taurine and indicine cattle macrophages: reactive oxygen species and potential outcomes to the development of immune response to infections

PONE-D-20-11705R1

Dear Dr. Wanessa Araujo Carvalho,

We’re pleased to inform you that your manuscript has been judged scientifically suitable for publication and will be formally accepted for publication once it meets all outstanding technical requirements.

Kind regards,

Gordon Langsley

Academic Editor

PLOS ONE

Additional Editor Comments (optional):

Reviewers' comments:

Reviewer's Responses to Questions

**Comments to the Author**

1. If the authors have adequately addressed your comments raised in a previous round of review and you feel that this manuscript is now acceptable for publication, you may indicate that here to bypass the “Comments to the Author” section, enter your conflict of interest statement in the “Confidential to Editor” section, and submit your "Accept" recommendation.

Reviewer #2: All comments have been addressed

2. Is the manuscript technically sound, and do the data support the conclusions?

Reviewer #2: Yes

3. Has the statistical analysis been performed appropriately and rigorously? 

Reviewer #2: Yes

4. Have the authors made all data underlying the findings in their manuscript fully available?

Reviewer #2: Yes

5. Is the manuscript presented in an intelligible fashion and written in standard English?

Reviewer #2: Yes

6. Review Comments to the Author

Reviewer #2: I thank the authors for the revised version of the manuscript and the new analysis. All my comments were addressed and I do not have any further question. I endorse the manuscript for publication.

7. PLOS authors have the option to publish the peer review history of their article (what does this mean?). If published, this will include your full peer review and any attached files.

Reviewer #2: **Yes: **Aude Remot

---

## [Editor Report · Acceptance letter]

27 Oct 2020

PONE-D-20-11705R1 

Lipopolysaccharide triggers different transcriptional signatures in taurine and indicine cattle macrophages: reactive oxygen species and potential outcomes to the development of immune response to infections 

Dear Dr. Carvalho:

I'm pleased to inform you that your manuscript has been deemed suitable for publication in PLOS ONE. Congratulations! Your manuscript is now with our production department. 

Kind regards, 

on behalf of

Dr. Gordon Langsley 

Academic Editor

PLOS ONE